# A transfer-aware, deployment-oriented evaluation framework for NetFlow-based intrusion detection systems (TAN-IDS)

Dung Ha Thanh*

Faculty of Information Technology, Saigon University, Ho Chi Minh, Vietnam

* htdung@sgu.edu.vn

## Abstract

Machine learning-based Intrusion Detection Systems (IDS) often report high detection accuracy under controlled, single-dataset evaluation, yet experience severe performance degradation when deployed in unseen network environments due to domain shift. To bridge this gap between laboratory benchmarking and real-world deployment, this paper presents TAN-IDS, a transfer-aware and deployment-oriented evaluation framework for NetFlow-based intrusion detection. Rather than proposing a new detection model, TAN-IDS contributes a methodological evaluation framework that unifies heterogeneous traffic datasets under a compact 8-dimensional NetFlow feature interface. This constrained representation supports interoperable and deployment-realistic evaluation across datasets collected in different network settings, enabling performance degradation to be more reliably attributed to domain shift rather than feature-space incompatibilities. Within this unified interface, TAN-IDS formalizes key deployment conditions as explicit evaluation scenarios, including in-dataset evaluation, direct cross-dataset transfer, mixed-domain training, and lightweight target-domain fine-tuning. Extensive experiments conducted within the proposed evaluation framework, using representative machine learning models and neural architectures, including a lightweight Transformer-based control model, show that strong in-dataset performance does not translate to cross-dataset robustness and that increased model complexity alone is insufficient to mitigate domain shift. In contrast, domain-aware training strategies are effective: mixed-domain training improves generalization, while fine-tuning with only 5% labeled target-domain data substantially recovers attack-class recall and F1-macro, exceeding 95% in several scenarios. Overall, TAN-IDS provides a reproducible, deployment-centric evaluation framework that reveals robustness limitations overlooked by benchmark-centric IDS evaluation.

**Data availability statement:** The datasets used in this study are publicly available. The UNSW-NB15 dataset is available from the Australian Centre for Cyber Security (https://research.unsw.edu.au/projects/unsw-nb15-dataset). The CIC-IDS2018 dataset is available from the Canadian Institute for Cybersecurity (https://www.unb.ca/cic/datasets/ids-2018.html). All data used in the experiments are derived from these publicly available datasets.

**Funding:** The author(s) received no specific funding for this work.

**Competing interests:** The authors have declared that no competing interests exist.

# 1. Introduction

Intrusion Detection Systems (IDS) remain a foundational cyber defense capability for identifying malicious activities and abnormal behaviors in network traffic. In operational environments, flow-based monitoring has become a preferred deployment interface, since payload inspection is increasingly constrained by encryption, privacy regulations, and high-throughput performance requirements [1,2]. In practice, IDS increasingly rely on machine learning (ML) and deep learning (DL) to learn detection rules from large-scale traffic data, and numerous surveys report consistently strong detection performance under controlled experimental settings [3,4]. However, a key deployment concern persists: much of the literature evaluates IDS under *closed-world* assumptions, where training and testing are performed on splits of the same dataset. As emphasized by Sommer and Paxson, such laboratory-oriented protocols can fail to reflect operational conditions in which modeling assumptions are routinely violated [5].

To isolate the impact of domain shift from confounding factors introduced by cross-dataset differences in attack label semantics, this study formulates intrusion detection as a binary classification task (*benign* vs. *attack*). Although multi-class detection is important in operational IDS, public benchmarks often provide incompatible attack taxonomies and label granularity, making cross-dataset label alignment unreliable. The binary formulation is thus adopted as a controlled evaluation setting to quantify cross-domain robustness and to avoid attributing performance changes to dataset-specific labeling artifacts rather than true distribution shift.

## 1.1. Why in-dataset evaluation is not enough

Empirical evidence increasingly shows that high in-dataset performance does not guarantee reliable detection after deployment. When a model trained on one dataset is applied to traffic collected in another environment, detection performance may drop sharply due to *dataset shift* (also referred to as *domain shift*). D'hooge *et al.* systematically quantified inter-dataset generalization and reported substantial degradation for supervised ML-based IDS under dataset transfer [6]. Apruzzese *et al.* further formalized cross-evaluation at scale, showing that cross-dataset results are often dramatically lower than in-dataset ones and warning against overly optimistic conclusions drawn from single-dataset benchmarks [7]. More recently, Cantone *et al.* confirmed that cross-dataset generalization remains a major bottleneck even for modern pipelines [8]. These findings motivate evaluation methodologies that treat cross-domain behavior as a first-class requirement for IDS research that aims to be application- and deployment-relevant [9]. This phenomenon has also been discussed in the context of training data selection and generalization strength [10].

## 1.2. Dataset heterogeneity and the feature-space barrier

A practical challenge in cross-dataset evaluation is that public IDS datasets are heterogeneous by construction: they differ in traffic composition, capture setups, labeling policies, and especially feature definitions [2]. For example, UNSW-NB15 was designed to reflect diverse attack families and benign behaviors in a laboratory

setting [11,12], while the CIC family of datasets targets contemporary attack scenarios with detailed flow statistics [13,14]. Even when both datasets are "flow-based", their native feature sets are often not directly comparable across datasets, creating a *feature-space barrier*: observed performance drops may conflate true domain shift with representation incompatibility. This barrier complicates fair comparison and undermines reproducibility across studies and tools.

### 1.3. Flow-based IDS and the case for NetFlow-standardization

From an operational perspective, flow-level monitoring (e.g., NetFlow/IPFIX) is widely adopted because it scales to high-throughput networks and remains viable under pervasive encryption. This motivates evaluation frameworks based on compact flow features that can be realistically extracted in production. Sarhan *et al.* advocated standardizing feature definitions for IDS datasets and argued that consistent feature sets are essential for reproducible comparisons [15,1]. Moreover, emerging flow benchmarks and telemetry datasets for IoT/IIoT further stress the need to evaluate detection tools across heterogeneous domains under comparable interfaces [16,17]. Therefore, a unified NetFlow representation can serve as a practical bridge between heterogeneous benchmarks and deployment-oriented evaluation.

### 1.4. Beyond accuracy: transfer-aware and explainable evaluation

Cross-domain assessment should not be limited to reporting degraded accuracy under dataset transfer, but should also support diagnosing *why* detection models fail and which mitigation strategies remain feasible under realistic deployment constraints. Prior work on explainable cross-domain evaluation has shown that post-hoc explanation techniques, such as LIME and SHAP, can reveal shifts in feature reliance and decision behavior when models are applied to unseen domains [18,19,20]. Such diagnostic tools provide valuable insight into failure modes under domain shift, but do not by themselves define how evaluation should be structured to reflect deployment conditions.

In parallel, transfer learning and domain adaptation offer principled mechanisms for mitigating performance loss under distribution shift, including fine-tuning with limited target-domain supervision and representation alignment strategies [21–23]. However, these techniques are often evaluated under ad-hoc or dataset-specific protocols, which makes it difficult to disentangle the effects of adaptation strategy, training data composition, and feature representation.

Recent Transformer-based flow IDS further demonstrate that increasing model capacity and representational expressiveness can improve in-dataset performance [24,25]. Nevertheless, model-centric advances alone do not eliminate the need for evaluation frameworks that explicitly organize *transfer-aware* scenarios under a consistent flow-level interface.

Accordingly, TAN-IDS positions explainability and transfer learning not as primary algorithmic contributions, but as complementary tools within a structured evaluation methodology. Additional relevant studies can be found in the literature [26,27,28,29,30,31,32].

In this work, the term *transfer-aware* refers to evaluation protocols that explicitly account for domain shift between training and deployment environments, including cross-dataset transfer, mixed-domain training, and limited-label target-domain adaptation.

By explicitly defining transfer-aware deployment scenarios under a unified NetFlow feature space, TAN-IDS enables systematic assessment of robustness, failure modes, and adaptation effectiveness beyond accuracy-centric reporting.

### 1.5. A gap in deployment-relevant evaluation methodology

Despite continuous advances in intrusion detection model architectures, the field still lacks lightweight and reproducible evaluation methodologies that can systematically isolate the impact of domain shift under consistent flow-level representations. Most existing studies emphasize in-dataset performance or rely on ad-hoc cross-dataset experiments. In such settings, differences in feature definitions, measurement semantics, and preprocessing can confound the interpretation of transfer performance. As a result, observed performance degradation is frequently attributable not only to domain shift, but also to uncontrolled feature incompatibilities.

To address this limitation, TAN-IDS shifts the focus from model-centric design to evaluation-centric methodology. Rather than proposing a new intrusion detection algorithm, this work concentrates on the design of evaluation protocols and benchmarking artifacts that enable systematic assessment of cross-domain robustness under deployment-relevant constraints. Central to this approach is a scenario controller that governs domain-aware data splitting, training composition, and adaptation strategies independently of the underlying classifier. This explicit separation allows diverse learning algorithms to be evaluated under identical and reproducible deployment assumptions, facilitating fair and comparable analysis across scenarios.

Although cross-dataset evaluation and NetFlow-based monitoring have been individually explored in prior work, TAN-IDS differs in how the evaluation problem is formulated and controlled. Specifically, deployment conditions are formalized as first-class experimental objects, and a unified flow-level feature interface is enforced across datasets. This formulation isolates the effects of domain shift under consistent measurement semantics, enabling transfer-aware and deployment-oriented assessment beyond conventional ad-hoc cross-dataset reporting. From a deployment perspective, the proposed 8-dimensional NetFlow representation is not merely a dimensionality reduction strategy. Instead, it reflects a deployment-oriented design choice that approximates a universal minimum interface for flow-based intrusion detection. The selected features correspond to flow-level statistics widely supported by contemporary routers, switches, and flow exporters. Importantly, they do not require deep packet inspection (DPI) or access to packet payloads. By constraining evaluation to this interoperable feature core, TAN-IDS targets cross-platform deployment scenarios where feature availability and measurement semantics vary across infrastructures. This design helps attribute performance degradation primarily to domain shift rather than feature incompatibility, narrowing the gap between laboratory benchmarking and real-world deployment.

## 1.6. The TAN-IDS framework

To address these limitations, we propose **TAN-IDS** (Transfer-Aware NetFlow-based IDS evaluation framework), a lightweight and reproducible framework for evaluating flow-based IDS under domain shift. TAN-IDS relies on a unified **8-dimensional NetFlow feature space** that can be consistently aligned across heterogeneous IDS datasets. In this work, the framework is instantiated using two widely adopted benchmarks—UNSW-NB15 and NF-CSE-CIC-IDS2018—which are mapped into the same compact NetFlow representation inspired by prior standardization efforts [15,1]. This shared representation enables direct cross-domain model comparison while reducing confounding factors caused by incompatible feature definitions.

Rather than treating cross-dataset testing as an auxiliary experiment, TAN-IDS formalizes deployment scenarios as first-class evaluation settings. The framework organizes in-dataset, cross-dataset, mixed-domain, and limited-label fine-tuning scenarios to reflect realistic IDS deployment conditions, shifting the focus from model-centric accuracy reporting to deployment-oriented robustness assessment under a unified NetFlow interface.

## 1.7. Evaluation scenarios and training strategies

Building on cross-evaluation findings in [6–8], TAN-IDS organizes experiments into domain-aware scenarios reflecting realistic deployment conditions. Specifically, we consider: (i) *in-dataset* evaluation, (ii) *cross-dataset* evaluation without adaptation, (iii) *mixed-domain* training across combined datasets, and (iv) *transfer-aware* evaluation via limited-label fine-tuning in the target domain. In this study, 5% labeled target-domain samples are used to simulate a realistic low-label adaptation scenario where only limited annotated traffic is available in a new deployment environment. These scenarios quantify both performance degradation caused by domain shift and the extent to which lightweight adaptation strategies can recover detection performance.

### 1.7.1. Contributions. This paper makes the following contributions:

- We propose TAN-IDS, a transfer-aware and deployment-oriented evaluation framework for flow-based intrusion detection systems, designed to assess robustness under realistic domain-shift scenarios.

- We introduce a scenario-driven evaluation protocol that explicitly captures key deployment settings, including in-dataset evaluation, cross-dataset transfer, mixed-domain training, and low-label target-domain fine-tuning.

- We define a unified NetFlow feature interface that enables fair and consistent cross-dataset evaluation across heterogeneous IDS benchmarks.

- Through systematic experiments within the proposed framework, we provide empirical evidence of the limitations of benchmark-centric IDS evaluation and assess the effectiveness of lightweight adaptation strategies for improving attack detection under domain shift.

**1.7.2. Paper organization.** The remainder of this paper is organized as follows. Section 2 reviews related work on cross-dataset IDS evaluation, flow-based feature standardization, and transfer-aware IDS. Section 3 presents the TAN-IDS framework and the unified NetFlow feature space. Section 4 describes the datasets, preprocessing, models, and evaluation protocol. Section 5 reports experimental results under all scenarios and discusses domain shift effects. Section 6 discusses threats to validity. Finally, Section 7 concludes the paper and outlines future research directions.

## 2. Related work

This section reviews prior work most relevant to TAN-IDS, focusing on cross-dataset evaluation of intrusion detection systems, flow-based feature representations and standardization, benchmark dataset heterogeneity, transfer learning for IDS, Transformer-based flow models, and explainability for cross-domain diagnosis. The review highlights both methodological advances and remaining gaps that motivate a transfer-aware, application-oriented evaluation framework.

### 2.1. Cross-dataset evaluation and inter-dataset generalization

A consistent conclusion across the IDS literature is that high in-dataset performance does not necessarily translate into reliable detection under deployment conditions. Early concerns regarding the limitations of laboratory evaluation were articulated by Sommer and Paxson, who argued that machine learning-based IDS often rely on implicit assumptions that are violated in real-world networks [5].

Subsequent empirical studies provided systematic evidence of this gap. D'hooge *et al* analyzed the inter-dataset generalization strength of supervised ML-based IDS and demonstrated that models trained on one dataset frequently exhibit substantial performance degradation when evaluated on another [6]. Apruzzese *et al* extended this line of work through a large-scale cross-evaluation study, showing that cross-dataset results are often dramatically lower than in-dataset ones and that conclusions drawn from single-dataset benchmarks can be overly optimistic [7]. More recently, Cantone *et al* confirmed that cross-dataset generalization remains a major bottleneck even for modern ML pipelines, reinforcing the need for evaluation protocols that go beyond closed-world assumptions [8].

While these studies convincingly quantify generalization loss, they primarily focus on reporting degradation, rather than on providing explicitly structured and reproducible evaluation frameworks that isolate domain shift under a consistent feature representation. TAN-IDS builds on these findings by organizing cross-dataset evaluation into explicit, transfer-aware scenarios grounded in a unified flow-level feature space.

### 2.2. Flow-based IDS and NetFlow feature standardization

Flow-based intrusion detection has gained increasing attention due to its scalability and compatibility with operational monitoring infrastructures. Surveys of IDS datasets emphasize that flow-level representations are more practical than packet payloads in environments constrained by encryption, privacy, and performance requirements [3,2].

Recent deep learning–based intrusion detection systems have also been developed for specialized IoT environments using flow or traffic statistics. For instance, HIDS-IoMT employs deep neural architectures to detect attacks in Internet of

Medical Things infrastructures [33]. Similarly, HIDS-RPL integrates hybrid deep learning techniques to identify routing-related attacks in RPL-based IoMT networks [34]. Beyond healthcare applications, deep learning IDS have also been explored in other vertical IoT domains such as smart agriculture, where hybrid architectures are used to enhance attack detection performance [35]. While these studies demonstrate promising results within specific application contexts, they primarily focus on improving detection architectures rather than establishing consistent feature representations or evaluating cross-dataset robustness.

Recognizing the challenge of dataset heterogeneity, Sarhan *et al* advocated the adoption of standardized feature sets for IDS datasets and highlighted NetFlow-style attributes as a practical basis for reproducible evaluation [15,1]. These efforts emphasize that inconsistent feature definitions across datasets constitute a major confounding factor in cross-dataset comparison.

However, existing standardization-oriented work largely focuses on feature-set guidance but does not explicitly address end-to-end evaluation methodology. In particular, prior studies do not systematically operationalize standardized flow features within structured cross-domain and transfer-aware evaluation scenarios. TAN-IDS addresses this gap by embedding a compact NetFlow feature space directly into a reproducible evaluation framework.

### 2.3. Benchmark datasets and dataset heterogeneity

Public IDS benchmarks remain essential for reproducible research, yet they differ substantially in traffic composition, attack scenarios, and data collection procedures. Classic benchmarks such as NSL-KDD further illustrate long-standing dataset limitations and distribution mismatch issues in IDS evaluation [36]. UNSW-NB15 was introduced to represent diverse modern attacks and benign behaviors in a controlled laboratory environment [11], with subsequent statistical analyses highlighting its characteristics and limitations [12]. The CIC family of datasets was designed to capture contemporary attack scenarios at scale, with detailed flow statistics and modern traffic profiles [13,14].

Beyond enterprise-style networks, datasets such as Bot-IoT and TON_IoT extend the scope to IoT and IIoT environments, further increasing dataset diversity [16,17]. Recent proposals for more realistic flow benchmarks, such as MAWI-Flow, underscore that realistic and deployment-oriented evaluation remains an open challenge [37]. Such efforts further emphasize the importance of realistic flow-level benchmarks for deployment-oriented IDS evaluation. In addition, the choice of training data itself has been shown to significantly influence generalization outcomes [10].

This diversity of datasets reinforces the need for evaluation frameworks that can compare IDS models across heterogeneous domains while minimizing representation-induced confounders. TAN-IDS directly addresses this need by aligning heterogeneous benchmarks within a unified NetFlow feature space.

### 2.4. Transfer learning and domain adaptation for IDS

When domain shift is unavoidable, transfer learning and domain adaptation provide principled tools to mitigate performance degradation. Foundational surveys categorize transfer learning paradigms such as fine-tuning, feature-based transfer, and representation alignment [21], while theoretical analyses formalize generalization bounds under distribution shift [23]. Domain-adversarial training further illustrates how representations can be encouraged to become domain-invariant [22], with related perspectives from invariant learning, unsupervised domain adaptation surveys, and distribution-matching criteria (e.g., MMD) [38,39].

In the IDS context, however, the primary challenge lies not only in developing adaptation algorithms, but also in evaluating them under consistent and deployment-relevant conditions. Many studies report improvements using transfer-oriented techniques under dataset-specific settings, yet evaluation is often performed using dataset-specific features or ad hoc protocols. TAN-IDS complements these approaches by providing a structured evaluation framework in which transfer learning strategies, such as fine-tuning with limited labeled target data, can be assessed under identical feature and scenario definitions.

## 2.5. Transformer-based IDS on flow data

Transformer architectures have achieved strong performance across a wide range of domains by leveraging self-attention mechanisms [25]. In flow-based intrusion detection, Manocchio *et al* proposed FlowTransformer, demonstrating that attention-based models can learn expressive representations from flow data and achieve competitive detection performance [24].

Despite these advances, Transformer-based IDS are predominantly evaluated under in-dataset settings, leaving open questions regarding robustness and transferability across datasets collected in different environments. This observation highlights that architectural sophistication alone does not obviate the need for transfer-aware evaluation protocols. TAN-IDS is designed to complement model-centric advances by providing an explicit evaluation lens for assessing Transformer and non-Transformer models under domain shift.

## 2.6. Explainability for cross-domain diagnosis

Explainable artificial intelligence (XAI) has been increasingly adopted to understand IDS decisions, audit feature reliance, and build trust in operational settings. Layeghy and Portmann specifically investigated explainable cross-domain evaluation for ML-based IDS, showing that explanation methods can reveal why detection performance degrades under domain shift [18]. Widely used post-hoc explanation techniques such as LIME and SHAP provide model-agnostic feature attributions [19,20], and broader surveys emphasize the importance of interpretability in security-critical ML systems [40,41], particularly in emerging industrial and operational settings [42].

While XAI-centric studies focus on interpretability as a primary objective, TAN-IDS positions explainability as a complementary diagnostic tool within a broader evaluation framework. By integrating explainability-oriented analysis into transfer-aware scenarios, TAN-IDS supports deeper understanding of cross-domain failures without conflating explanation with detection model design.

## 2.7. Summary and positioning of TAN-IDS

Prior work has established that cross-dataset generalization is a fundamental challenge for ML-based IDS and that in-dataset evaluation can substantially overestimate deployment performance [6–8]. At the same time, dataset heterogeneity and incompatible feature definitions hinder fair comparison and reproducibility, motivating standardization efforts for flow-based IDS [15,1]. Transfer learning and explainability offer valuable tools for mitigating and diagnosing domain shift, while recent Transformer-based approaches highlight the need for evaluation protocols that extend beyond closed-world testing.

Recent deep learning-based IDS architectures have also been proposed for specialized IoT environments, including Internet of Medical Things (IoMT) networks and other domain-specific deployments [33–35]. Although these studies demonstrate promising detection performance within specific application contexts, they primarily focus on improving model architectures rather than establishing reproducible evaluation frameworks that systematically assess cross-dataset robustness and deployment-oriented generalization.

In contrast to predominantly model-centric or dataset-specific studies, TAN-IDS introduces a lightweight, reproducible, and application-oriented evaluation framework that unifies heterogeneous datasets into a compact NetFlow feature space and explicitly organizes in-dataset, cross-dataset, mixed-domain, and transfer-aware evaluation scenarios. This positioning enables systematic and fair assessment of IDS robustness under realistic domain shift conditions.

To position TAN-IDS in this landscape, Table 1 summarizes representative studies, their datasets, feature levels, evaluation settings, and limitations relevant to transfer-aware assessment. Complementarily, Table 2 provides a compact matrix view of key capabilities across prior work and TAN-IDS. In contrast to model-centric approaches or dataset-specific evaluations, TAN-IDS contributes a *lightweight, reproducible, transfer-aware* evaluation framework centered on a unified NetFlow feature space and an explicit suite of in-dataset, cross-dataset, mixed-domain, and fine-tuning scenarios.

**Table 1. Representative related work and positioning of TAN-IDS.**

| Ref. | Dataset(s) | Feature level | Method / focus | Evaluation | Scope relative to TAN-IDS |
|---|---|---|---|---|---|
| [6] | Multiple IDS datasets | Tabular (varies) | Inter-dataset generalization | In-dataset / Cross-dataset | Does not focus on flow-level or NetFlow-aligned feature standardization and does not explicitly consider structured transfer-aware deployment scenarios |
| [7] | Multiple IDS datasets | Tabular (varies) | Large-scale cross-evaluation | Cross-dataset | Cross-dataset evaluation is performed under heterogeneous feature spaces; a unified flow-level interface is not the primary focus |
| [8] | Multiple IDS datasets | Tabular (varies) | Cross-dataset generalization | Cross-dataset | Does not target unified NetFlow feature alignment or explicitly scenario-driven transfer evaluation |
| [18] | UNSW/CIC/others | Flow/header statistics | Explainable cross-domain evaluation | Cross-dataset + XAI | XAI-centric analysis; does not explicitly consider lightweight mixed-domain or limited-label fine-tuning scenarios |
| [15,1] | NetFlow datasets (survey/ standardization) | Flow/ NetFlow | Standard feature-set guidance | Conceptual benchmarks | Focuses on feature standardization and dataset characterization; not designed as an end-to-end, scenario-driven transfer evaluation framework across UNSW–CIC |
| [24] | Flow-based IDS datasets | Flow/ NetFlow | Transformer-based IDS modeling | Mostly in-dataset | Model-centric design; cross-dataset and transfer-aware deployment evaluation are not the primary focus |
| **TAN-IDS (this work)** | UNSW-NB15 & NF-CSE-CIC-IDS2018 | **Unified 8-D NetFlow** | **Transfer-aware evaluation framework** | **In-, cross-, mixed, FT** | **Isolates domain shift under a consistent NetFlow interface using deployment-oriented, structured evaluation scenarios** |

"Unified 8-D NetFlow" denotes a fixed NetFlow-aligned feature interface instantiated consistently across datasets. "FT" indicates limited-label fine-tuning in the target domain.

**Table 2. Related work matrix comparing evaluation capabilities across representative IDS studies.**

| Work | Cross | Flow | Unified | TA | XAI |
|---|---|---|---|---|---|
| [6] | ✓ | – | – | – | – |
| [7] | ✓ | – | – | – | – |
| [8] | ✓ | – | – | – | – |
| [18] | ✓ | ✓ | – | – | ✓ |
| [15,1] | – | ✓ | ✓ | – | – |
| [24] | – | ✓ | – | – | – |
| **TAN-IDS (this work)** | ✓ | ✓ | ✓ | ✓ | ✓† |

**Cross**: cross-dataset evaluation; **Flow**: flow-based traffic features; **Unified**: a fixed NetFlow-aligned feature interface consistently instantiated across datasets; **TA**: explicitly scenario-driven transfer protocols, including mixed-domain training and limited-label fine-tuning; **XAI**: explainability for diagnostic analysis.

† TAN-IDS supports post-hoc explanation tools (e.g., LIME/SHAP) for diagnostic purposes; explainability is not a core contribution of the framework.

## 3. The Proposed TAN-IDS Framework

### 3.1. Framework perspective

Rather than proposing a new intrusion detection model, this work formulates TAN-IDS as a structured, transfer-aware evaluation framework for flow-based intrusion detection systems. The framework maps datasets, deployment scenarios, learning models, and performance metrics into a unified assessment pipeline, with the explicit goal of evaluating deployment readiness under domain shift. Although our empirical study instantiates TAN-IDS using two representative public datasets, the framework itself is dataset-agnostic and can be readily extended to additional NetFlow-based datasets.

This section presents **TAN-IDS**, a *Transfer-Aware NetFlow-based Intrusion Detection System* evaluation framework designed to systematically assess the generalization behavior of IDS models across heterogeneous network environments. TAN-IDS does not introduce a new detection algorithm; instead, it provides a lightweight and reproducible evaluation framework that (i) aligns heterogeneous flow-based datasets into a unified NetFlow feature space and (ii) organizes domain-aware training and testing scenarios that reflect realistic deployment conditions.

Fig 1 provides a high-level overview of the TAN-IDS framework. It explicitly separates feature alignment, model training, and scenario-driven evaluation under different domain assumptions, enabling reproducible and transfer-aware analysis of IDS performance beyond conventional benchmark-centric evaluations.

## 3.2. Design goals and scope

TAN-IDS is designed with the following goals in mind: (i) *Transfer awareness*: explicitly expose performance degradation when models are transferred across datasets collected in different environments; (ii) *Flow-level realism*: rely on compact NetFlow-style features that are feasible to extract in operational networks; (iii) *Model agnosticism*: support a wide range of ML and DL classifiers without imposing architecture-specific constraints; and (iv) *Reproducibility*: minimize ad-hoc preprocessing choices by using a unified feature definition and standardized evaluation scenarios.

Accordingly, TAN-IDS focuses on flow-based IDS and cross-dataset evaluation between widely used benchmarks rather than proposing new detection architectures or dataset-specific feature engineering pipelines.

## 3.3. Unified NetFlow feature space

A central component of TAN-IDS is a **unified 8-dimensional NetFlow feature space**. This representation can be consistently extracted from heterogeneous IDS datasets. This design is inspired by standardization efforts in flow-based intrusion detection and NetFlow dataset analysis, which emphasize compact, interpretable, and widely supported flow attributes.

The choice of an 8-dimensional representation is motivated by deployment-oriented considerations. The selected features correspond to core flow-level statistics that are consistently supported by common NetFlow exporters and monitoring infrastructures. By restricting the representation to this compact and interoperable feature set, TAN-IDS reduces dataset-specific feature dependencies and enables fair cross-dataset comparison under a unified flow-level interface.

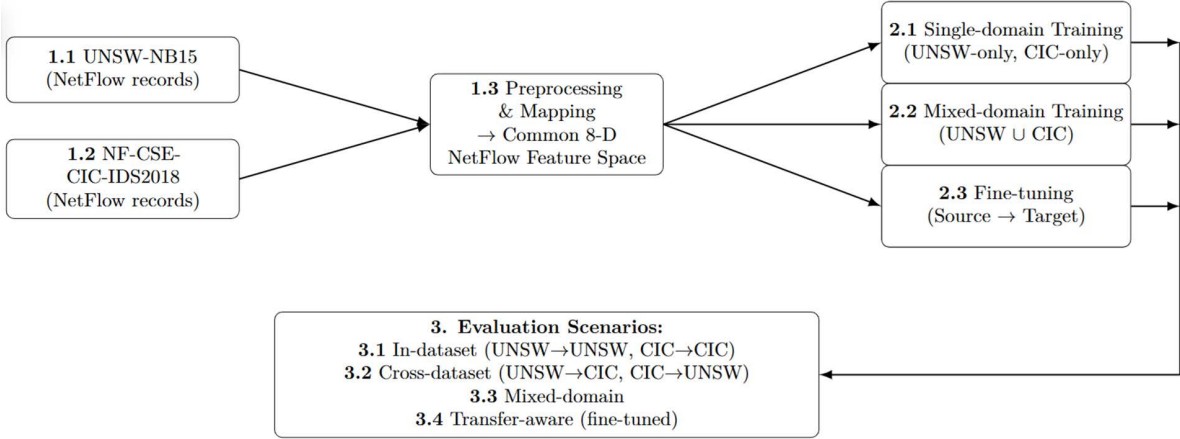

**Fig 1. Overview of the proposed TAN-IDS framework.** TAN-IDS unifies heterogeneous IDS datasets into a common NetFlow feature space and organizes domain-aware evaluation scenarios, including in-dataset, cross-dataset, mixed-domain, and transfer-aware fine-tuning.

Let each network flow be represented as a vector

$$\mathbf{x} = [f_1, f_2, \ldots, f_8] \in \mathbb{R}^8 \tag{1}$$

where the selected features capture essential temporal, volumetric, and directional properties of network communication, such as flow duration, packet counts, byte counts, and packet rate statistics. These features are available or derivable in both UNSW-NB15 and NF-CSE-CIC-IDS2018, enabling direct alignment without relying on dataset-specific or high-dimensional attributes.

This design choice aligns with prior efforts on standardized flow-level feature representations for intrusion detection [1,15].

By projecting all flows into this shared NetFlow space, TAN-IDS reduces structural discrepancies between datasets and allows observed performance differences to be more confidently attributed to *domain shift* rather than incompatible feature definitions. This feature alignment step corresponds to the leftmost block in Figs 1 and 2.

The eight-dimensional feature space is not claimed to be information-maximizing. Instead, it is a deployment-driven intersection of flow attributes that can be consistently derived across heterogeneous datasets and monitoring pipelines. This design prioritizes portability and comparability: by fixing a unified NetFlow-like interface, TAN-IDS reduces dataset-specific feature artifacts and enables fair transfer-aware evaluation under domain shift.

This choice represents a deliberate trade-off between representational richness and cross-dataset compatibility. While richer feature sets may benefit within-dataset accuracy, they often reduce portability and can confound transfer results due to inconsistent feature definitions across datasets.

### 3.4. Datasets and domain alignment

In this work, TAN-IDS instantiates the unified NetFlow representation on two widely adopted IDS benchmarks: UNSW-NB15 and NF-CSE-CIC-IDS2018. Although both datasets are flow-based, they differ substantially in traffic composition, attack distributions, capture environments, and labeling policies. These differences make them suitable representatives of distinct domains for cross-dataset evaluation.

Let $\mathcal{D}_s$ and $\mathcal{D}_t$ denote the source and target domains, with joint data–label distributions $P_s(\mathbf{x}, y)$ and $P_t(\mathbf{x}, y)$, respectively. Cross-dataset evaluation in TAN-IDS explicitly considers the case where

$$P_s(\mathbf{x}, y) \neq P_t(\mathbf{x}, y), \tag{2}$$

which reflects realistic deployment conditions under domain shift.

The alignment process consists of: (i) extracting or deriving the selected NetFlow features from each dataset, (ii) applying consistent preprocessing steps, including normalization, and (iii) harmonizing class labels to a common attack/benign taxonomy. Importantly, no dataset-specific feature augmentation is introduced, ensuring that all models operate on the same information content across domains.

### 3.5. Transfer-aware evaluation scenarios

**Scenario controller** Within TAN-IDS, the scenario controller is a conceptual evaluation component that explicitly configures domain-aware train/test splits and adaptation protocols corresponding to in-dataset, cross-dataset, mixed-domain, and fine-tuning scenarios. Crucially, the controller operates independently of the classifier architecture, ensuring that all models are evaluated under identical, reproducible, and deployment-consistent conditions.

To systematically characterize generalization behavior under domain shift, TAN-IDS organizes experiments into four complementary *domain-aware evaluation scenarios*, which are illustrated conceptually in the framework pipeline.

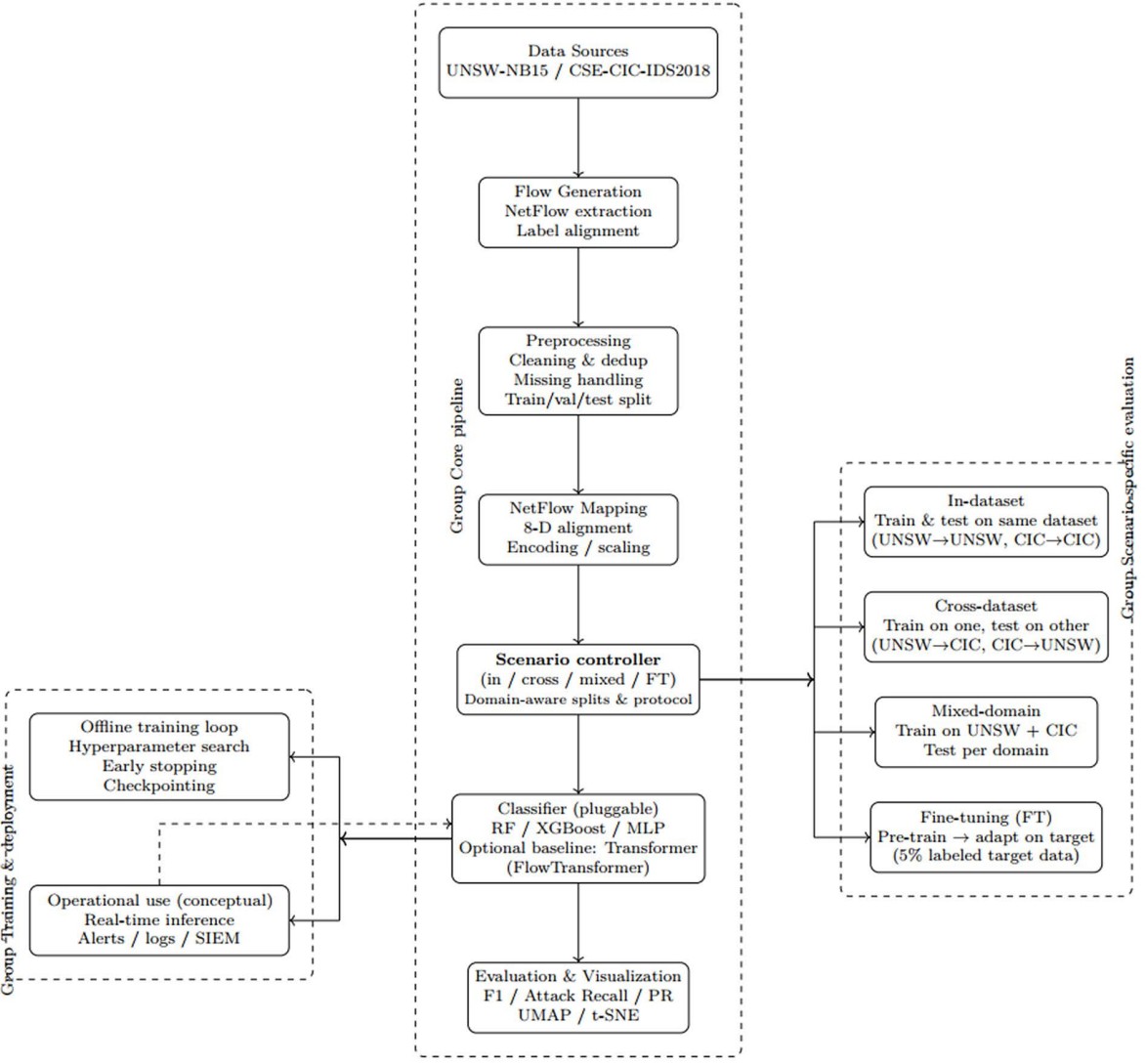

**Fig 2. Detailed pipeline of the TAN-IDS evaluation framework.** The scenario controller defines domain-aware evaluation protocols, including in-dataset, cross-dataset, mixed-domain, and transfer-aware fine-tuning, independently of the classifier choice. Pluggable classifiers (Random Forest, XGBoost, Multilayer Perceptron, with an optional Transformer-based baseline) are trained and evaluated within a unified 8-dimensional NetFlow feature space. All preprocessing statistics are computed exclusively on training splits to prevent data leakage.

Given a classifier $f_\theta$ trained under a specific evaluation scenario $S$, its generalization performance on a target domain $\mathcal{D}_t$ is assessed through the expected target-domain risk:

$$\mathcal{R}_t(f_\theta) = \mathbb{E}_{(\mathbf{x},y) \sim P_t} \left[ \ell\big(f_\theta(\mathbf{x}), y\big) \right],$$ (3)

where $P_t(\mathbf{x}, y)$ denotes the joint data distribution of the target domain and $\ell(\cdot)$ is a task-specific loss function.

This formulation provides a common analytical lens for comparing how different training strategies and deployment scenarios influence performance degradation and recovery under domain shift, independent of the underlying model design.

**3.5.1. In-dataset evaluation.** Models are trained and tested on disjoint splits of the same dataset. This scenario corresponds to the dominant evaluation protocol in the IDS literature and serves as a reference for upper-bound performance under closed-world assumptions, where training and deployment data are assumed to follow the same distribution.

**3.5.2. Cross-dataset evaluation.** Models are trained on one dataset (source domain) and directly evaluated on another dataset (target domain) without any form of adaptation. This scenario isolates the raw effect of domain shift and quantifies the generalization loss that occurs when deployment conditions differ from the training environment.

**3.5.3. Mixed-domain training.** Training data from multiple domains are combined to learn a single model, which is then evaluated separately on each domain. This scenario assesses whether exposure to heterogeneous traffic during training improves robustness and generalization across distinct network environments.

**3.5.4. Transfer-aware fine-tuning.** A model pre-trained on a source domain is adapted to a target domain using a limited amount of labeled target-domain data. This scenario reflects practical deployment settings in which only small labeled samples from a new environment may be available and evaluates the effectiveness of lightweight transfer learning strategies.

In this scenario, model parameters are initialized from a source-trained solution $\theta_s$ and updated using a limited labeled target dataset $\mathcal{D}_t^{(l)}$:

$$\theta_t = \arg\min_{\theta} \mathbb{E}_{(\mathbf{x},y) \sim \mathcal{D}_t^{(l)}} \left[ \ell(f_\theta(\mathbf{x}), y) \right], \tag{4}$$

where $\ell(\cdot)$ denotes the task-specific loss function. This formulation follows standard transfer learning paradigms widely studied in the literature [21].

Together, the four evaluation scenarios enable a structured comparison of performance degradation and recovery under different levels of domain awareness.

As illustrated in Fig 2, the TAN-IDS pipeline consists of four main stages. First, raw traffic records from heterogeneous datasets are mapped into a unified NetFlow feature space using consistent extraction and preprocessing steps. Second, datasets are aligned and split according to the selected evaluation scenario. Third, models are trained using one of the supported strategies, including single-domain training, mixed-domain training, or transfer-aware fine-tuning. Finally, trained models are evaluated under domain-aware testing conditions, and both quantitative metrics and qualitative analyses are produced to assess generalization behavior.

Table 3 summarizes the evaluation scenarios supported by TAN-IDS and clarifies the role of each scenario in exposing performance degradation and recovery under domain shift.

The four evaluation scenarios are visually summarized in Fig 1 and operationalized through the pipeline shown in Fig 2.

**Algorithm 1 TAN-IDS: Transfer-Aware NetFlow-based IDS Evaluation Pipeline**

**Require:** Datasets $\mathcal{D} = \{\mathcal{D}_1, \mathcal{D}_2, \dots\}$, NetFlow feature extractor $\Phi(\cdot)$, Classifier family $\mathcal{M}$, Evaluation scenario $S$
**Ensure:** Performance metrics under scenario $S$
1: **for** each dataset $\mathcal{D}_i \in \mathcal{D}$ **do**
2:  Extract flow records and compute NetFlow features: $\mathbf{X}_i \leftarrow \Phi(\mathcal{D}_i)$
3:  Apply consistent preprocessing and normalization
4: **end for**
5: Align datasets into unified feature space $\mathbb{R}^8$
6: Split training and test sets according to scenario $S$
7: **if** $S$=Mixed-domain **then**
8:  Merge training data from all source domains
9: **end if**
10: Initialize model $m \in \mathcal{M}$

**Table 3. Transfer-aware evaluation scenarios supported by TAN-IDS.**

| Scenario | Training data | Test data | Purpose |
|----------|---------------|-----------|---------|
| In-dataset | Single dataset | Same dataset | Reference performance under closed-world assumptions |
| Cross-dataset | Source dataset | Target dataset | Quantify performance degradation caused by domain shift |
| Mixed-domain | Union of multiple datasets | Each dataset separately | Assess robustness gained from heterogeneous training data |
| Transfer-aware fine-tuning | Source dataset with limited target samples | Target dataset | Evaluate performance recovery under limited target-domain supervision |

```
11: Train m on selected training data
12: if S=Transfer-aware fine-tuning then
13:   Fine-tune m using limited labeled target-domain samples
14: end if
15: Evaluate m on target test data
16: Compute detection metrics and optional explainability outputs
17:   return Evaluation results under scenario S
```

Algorithm 1 formalizes the TAN-IDS pipeline and highlights how unified feature alignment and scenario-aware training are combined to enable reproducible cross-domain evaluation.

### 3.6. Model-agnostic framework and supported classifiers

Building upon the pipeline shown in Fig 2, this subsection discusses the model-agnostic nature of TAN-IDS and the classifiers instantiated in this study. TAN-IDS is explicitly model-agnostic and can, in principle, be instantiated with a wide range of classical machine learning models, ensemble methods, and deep learning architectures for flow-based intrusion detection.

In this work, the framework is demonstrated using representative classifiers, including Random Forest, XGBoost, and Multilayer Perceptron, which cover a spectrum of learning biases and model complexities commonly used for tabular flow data. Random Forest [43], XGBoost [44], and Multilayer Perceptron serve as strong and widely adopted baselines in IDS research.

Importantly, TAN-IDS does not restrict the choice of classifier and can incorporate more advanced architectures when desired. To this end, a lightweight Transformer-based baseline is included as a capacity-rich control model rather than a claim of architectural superiority. Its purpose is to examine whether increased model capacity and self-attention alone are sufficient to improve robustness under domain shift within a unified NetFlow feature space. A lightweight configuration is intentionally used to control computational cost and to isolate the effect of evaluation scenarios rather than model scale.

By decoupling the evaluation protocol from specific model designs, TAN-IDS enables fair and consistent comparison across methods under identical domain-aware conditions.

### 3.7. Framework output and analysis

The output of TAN-IDS consists of both quantitative and qualitative assessments. Quantitatively, standard detection metrics (e.g., accuracy, precision, recall, and F1-score) are reported consistently across all scenarios to enable direct comparison. Qualitatively, dimensionality-reduction techniques such as UMAP and t-SNE can be applied to the unified feature space or learned representations to visualize domain gaps and representation alignment under different training strategies [45,46].

By combining these perspectives, TAN-IDS provides insight not only into *how much* performance degrades under domain shift, but also into *how* models and feature representations behave when transferred across domains.

### 3.8. Summary

In summary, TAN-IDS provides a transfer-aware evaluation framework for flow-based intrusion detection that unifies heterogeneous datasets into a compact NetFlow feature space and organizes realistic domain-aware evaluation scenarios. Rather than proposing a new detection algorithm, TAN-IDS is intended to complement existing and future IDS models by offering a principled, reproducible, and deployment-oriented lens through which cross-domain generalization can be systematically studied.

## 4. Experimental setup

This section describes the datasets, preprocessing pipeline, training and evaluation protocols, model configurations, and evaluation metrics used to assess the proposed TAN-IDS framework. The experimental design aims to ensure fairness, reproducibility, and a systematic analysis of cross-dataset generalization under domain shift.

### 4.1. Datasets and preprocessing

We conduct experiments on two widely used public intrusion detection datasets constructed at the flow level:

- **UNSW-NB15**: A dataset generated at the Cyber Range Lab, University of New South Wales, consisting of approximately 2.5 million network flows covering normal traffic and nine attack families. The dataset combines real background traffic with synthetic attack scenarios, resulting in diverse traffic patterns.

- **NF-CSE-CIC-IDS2018**: A NetFlow-based version of the CIC-IDS2018 dataset released by the Canadian Institute for Cybersecurity (CIC), consisting of approximately 8.39 million flow records. Each record represents a network flow aggregated in a NetFlow/IPFIX-style format, and the resulting traffic characteristics differ substantially from UNSW-NB15.

Due to the structural heterogeneity between the two datasets, we do not directly use their original feature spaces. Instead, both datasets are mapped into a **shared NetFlow feature space with eight dimensions**, as detailed in Section 3. This unified representation enables direct cross-dataset evaluation while preserving flow-level semantics commonly available in real-world deployments.

The preprocessing pipeline consists of the following steps:

1. Removal of corrupted or incomplete flows (e.g., missing duration or packet statistics).

2. **Binary label normalization**, where all attack categories are merged into a single *attack* class and normal traffic is labeled as *benign*. This choice is made to avoid confounding effects introduced by cross-dataset differences in attack taxonomies, label granularity, and semantic mismatches; consequently, the evaluation is better isolated to quantify domain shift under a consistent labeling scheme.

3. Feature normalization using **standard scaling** to mitigate scale discrepancies among NetFlow attributes.

Accordingly, the binary formulation in TAN-IDS should be interpreted as a controlled evaluation setting for cross-domain robustness, rather than a claim that fine-grained attack classification is unnecessary in operational IDS.

All flow duration values are converted to a common unit (milliseconds) prior to feature normalization to ensure unit consistency across datasets.

Table 4 reports summary statistics of the two datasets after preprocessing and feature alignment, including the number of flows, class distribution, attack ratio, and feature dimensionality. Table 5 details the one-to-one mapping used to align UNSW-NB15 and NF-CSE-CIC-IDS2018 into the unified 8-D NetFlow feature space.

**Table 4. Statistics of the NetFlow datasets used in TAN-IDS.**

| Dataset | Symbol | #Flows | #Benign | #Attack | Attack ratio (%) | #Features |
|---|---|---|---|---|---|---|
| UNSW-NB15 (NetFlow) | UNSW | 2,539,739 | 2,218,456 | 321,283 | 12.65 | 8 |
| NF-CSE-CIC-IDS2018 (NetFlow) | CIC | 8,392,401 | 7,373,198 | 1,019,203 | 12.14 | 8 |

**Table 5. Mapping and description of the eight common NetFlow features between UNSW-NB15 (NetFlow) and NF-CSE-CIC-IDS2018. Non-continuous fields (proto, and tcp lags) are encoded deterministically; ports are treated as bounded numeric variables.**

| Common attribute | UNSW | CIC | Description |
|---|---|---|---|
| flow_duration | dur | Flow_Duration | Flow duration converted to milliseconds during processing to ensure unit consistency across datasets. |
| pkt_total | pkts | Tot_Fwd_Pkts + Tot_Bwd_Pkts | Total number of packets in the flow (forward + backward). |
| byte_total | bytes | Tot_Fwd_Bytes + Tot_Bwd_Bytes | Total number of bytes in the flow (forward + backward). |
| src_port | sport | Src_Port | Source port (transport layer). |
| dst_port | dport | Dst_Port | Destination port (transport layer). |
| proto | proto | Protocol | Transport protocol (TCP, UDP, ICMP, ...). |
| tcp_flags | state | Fwd_Pkt_Flags / Bwd_Pkt_Flags | Aggregated TCP flag bits for the flow. |
| packets_per_sec | pkts_rate | Flow_Pkts_s | Packet rate (packets per second). |

Rate-based temporal features, such as packets per second, are inherently time-normalized and therefore directly comparable across datasets without additional unit conversion.

**4.1.1. Encoding and unit harmonization for the 8-D NetFlow space.** To ensure a consistent numeric representation across UNSW-NB15 (NetFlow) and NF-CSE-CIC-IDS2018, we harmonize units and apply deterministic encodings to non-continuous fields. First, flow_duration is converted to a common unit (milliseconds) prior to normalization. Second, pkt_total and byte_total are computed as the sum of forward and backward statistics in NF-CSE-CIC-IDS2018 (Tot_Fwd_Pkts+Tot_Bwd_Pkts, Tot_Fwd_Bytes+Tot_Bwd_Bytes) and mapped to their UNSW counterparts (pkts, bytes). Third, transport-layer ports (src_port, dst_port) are treated as bounded numeric variables in [0,65535] and standardized together with other features using training-split statistics. The protocol field (proto) is encoded via a fixed dictionary shared across datasets (e.g., TCP, UDP, ICMP, ... mapped to stable integer IDs) to preserve the compact 8-D setting without expanding dimensionality. Finally, tcp_flags is represented as a single numeric bit-mask: for NF-CSE-CIC-IDS2018, forward/backward packet flag fields are aggregated using a bitwise OR; for UNSW-NB15, the state field is encoded deterministically using a fixed dictionary shared across experiments to preserve reproducibility. Standard scaling is fitted on the training split only and then applied unchanged to validation and test splits to prevent information leakage.

We do not use raw IP addresses as model inputs, as they can introduce dataset-specific identity leakage and do not generalize across deployments. Source and destination ports are consistently treated as bounded numeric flow attributes in all experiments reported in this paper, without categorical grouping. Continuous features are normalized using the same protocol across datasets.

## 4.2. Data splitting and evaluation protocol

For each dataset, the preprocessed data are divided into training, validation, and test subsets using a **stratified splitting strategy** to preserve the benign/attack ratio:

• 70% for training,

- 15% for validation,

- 15% for testing.

All splits are generated with a fixed random seed to ensure reproducibility. Based on these splits, we define four evaluation scenarios:

- **In-dataset evaluation**: Training and testing are performed on the same dataset (UNSW→UNSW, CIC→CIC).

- **Cross-dataset evaluation**: Models are trained on the source dataset and directly tested on the target dataset without using any labeled samples from the target domain during training (UNSW→CIC, CIC→UNSW).

- **Fine-tuning**: Models are first trained on the source domain and then adapted using a fixed budget of **5% labeled samples** from the target domain, after which evaluation is performed on the target test set.

- **Mixed-domain training**: Training data from both datasets are combined to form a mixed-domain training set, while testing is conducted separately on each dataset.

**4.2.1. Fine-tuning budget and sampling strategy.** In the transfer-aware fine-tuning scenario, models pre-trained on a source domain are adapted using a fixed budget of **5% labeled samples** from the target domain. The target-domain subset is selected via stratified random sampling with respect to the binary class label (benign/attack) to preserve class proportions. This budget is chosen as a representative low-label regime that reflects practical deployment settings, where only a small amount of labeled data from the new environment is typically available.

Under this setting, fine-tuning uses approximately $0.05|\mathcal{D}_t|$ labeled flows from the target dataset, corresponding to several tens of thousands of samples for NF-CSE-CIC-IDS2018 and proportionally fewer samples for UNSW-NB15.

**4.2.2. Mixed-domain data composition.** For mixed-domain training, we concatenate UNSW-NB15 and NF-CSE-CIC-IDS2018 using their natural sample proportions (i.e., without domain-level downsampling). The mixed-domain training split is stratified by the binary class label (benign/attack) to preserve class distribution. As a result, the mixed-domain training set is dominated by the larger dataset (NF-CSE-CIC-IDS2018); we keep this choice to reflect a large-scale monitoring setting and to avoid introducing additional sampling heuristics that may confound the evaluation.

This protocol enables a systematic investigation of domain shift effects and the effectiveness of different mitigation strategies under controlled and reproducible deployment-oriented conditions.

## 4.3. Models and hyperparameter configuration

We evaluate TAN-IDS using a diverse set of machine learning and deep learning models commonly employed in flow-based intrusion detection:

- **Random Forest (RF)**: An ensemble of decision trees offering robustness to noise and class imbalance. Class imbalance is a well-known challenge in intrusion detection evaluation [47]. While various remedies such as resampling methods (e.g., SMOTE [48]) and loss re-weighting have been proposed in the literature, we do not apply dataset-specific imbalance handling techniques in TAN-IDS to avoid introducing additional confounding factors across domains.

- **XGBoost (XGB)**: A gradient boosting model known for strong performance on tabular data.

- **Multilayer Perceptron (MLP)**: A feed-forward neural network serving as a deep learning baseline for the 8-dimensional NetFlow feature space.

In addition, we include a lightweight Transformer-based baseline, denoted as FlowTransformer-lite (hereafter referred to as FlowTrans), which applies self-attention mechanisms to the unified NetFlow features.

All models are trained using fixed hyperparameter configurations across all experimental scenarios to ensure fair comparisons. The detailed hyperparameter settings are summarized in Table 6. For neural network–based models (MLP and FlowTransformer-lite), optimization is performed using the Adam optimizer [49], with dropout regularization to mitigate overfitting [50,51]. For highly imbalanced settings, loss re-weighting variants such as focal loss have been proposed in prior work [52]; however, they are not applied in our experiments to maintain consistent training protocols across all datasets and evaluation scenarios.

## 4.4. Evaluation metrics

Model performance is assessed using standard metrics for intrusion detection:

- **Accuracy**,
- **Precision** (attack class),
- **Recall** (attack class),
- **Macro-averaged F1-score**,
- **Confusion matrix** (for error analysis).

For cross-dataset scenarios, we emphasize **attack-class recall** and **macro-averaged F1-score**, as these metrics more accurately reflect the risk of missed attacks under domain shift conditions. Precision–recall metrics are more informative than ROC analysis under class imbalance [53,54].

## 4.5. Experimental environment and reproducibility

All experiments are conducted on a standard server environment equipped with multi-core CPUs and 32 GB RAM. The implementation is based on Python 3.10, using scikit-learn, XGBoost, and TensorFlow 2.x.

To ensure reproducibility, the entire experimental workflow follows an **artifact-based design**. Intermediate results, including preprocessed datasets, scalers, trained models, predictions, and evaluation tables, are persistently stored. This design enables individual experiments to be rerun or extended without retraining models from scratch, supporting transparent and repeatable research.

**4.5.1. Reproducibility considerations.** TAN-IDS is designed with reproducibility as a primary objective, following a modular evaluation pipeline that explicitly decouples dataset adaptation, scenario definition, model training, and metric

**Table 6. Key hyperparameters for MLP, Random Forest, XGBoost, and FlowTransformer-lite (baseline) in TAN-IDS.**

| Model | Key hyperparameter settings |
|---|---|
| MLP | Three hidden layers (128–64–32) with ReLU activation and dropout rate of 0.3; Adam optimizer with learning rate $10^{-3}$; batch size 512; 20 training epochs. |
| Random Forest | Number of trees ($n_{estimators}$=200); unrestricted maximum depth; parallel training enabled ($n_{jobs}$=−1); random seed fixed to 42. |
| XGBoost | $n_{estimators}$=300; maximum depth 8; learning rate 0.1; subsample ratio 0.8; column subsampling ratio 0.8; objective function binary:logistic; histogram-based tree construction; evaluation metric logloss; parallel training enabled. |
| FlowTransformer-lite (baseline) | Model dimension $d_{model}$=64; four attention heads; feed-forward dimension 128; two Transformer layers; dropout rate 0.1; Adam optimizer with learning rate $10^{-3}$; batch size 512; 20 training epochs. |

computation. This design enables consistent instantiation of identical evaluation scenarios across different datasets and learning algorithms, supporting systematic and transfer-aware analysis of IDS performance under controlled deployment assumptions. To ensure a fair and deployment-oriented evaluation of cross-domain robustness, hyperparameters for all models were fixed across datasets and evaluation scenarios, thereby isolating the effects of domain shift and training data composition rather than conflating transfer performance with dataset-specific hyperparameter optimization. Except for explicitly defined fine-tuning scenarios, no target-domain information was used for model selection or hyperparameter adjustment. The implementation code will be publicly released upon acceptance.

## 5. Results and discussion

This section presents and analyzes the experimental results of TAN-IDS across multiple deployment-relevant evaluation scenarios, including in-dataset, cross-dataset, mixed-domain training, and lightweight fine-tuning. The discussion focuses on the effects of domain shift, the comparative behavior of representative learning models, and the effectiveness of mitigation strategies under realistic deployment constraints.

Given the inherent class imbalance in network intrusion detection datasets, overall accuracy alone is insufficient to characterize deployment-relevant performance. Accordingly, the analysis emphasizes attack recall, macro-averaged F1-score, and balanced accuracy to ensure that observed performance trends are not driven by bias toward the dominant normal class. These metrics better reflect the operational objectives of intrusion detection systems, where missed attacks typically incur higher costs than false alarms.

### 5.1. In-dataset performance

In-dataset results are primarily analyzed in terms of attack recall and macro-averaged F1-score, as these metrics are more robust to class imbalance and better reflect deployment-relevant performance. Fig 3(a–b) report in-dataset performance in terms of macro-F1 and attack recall, respectively.

Tree-based models (Random Forest and XGBoost) consistently deliver high macro-averaged F1-scores, reflecting their robustness on tabular NetFlow features. The MLP baseline achieves competitive performance, demonstrating that the proposed 8-dimensional NetFlow feature space preserves sufficient discriminative information for deep learning models. Table 7 summarizes the corresponding in-dataset performance metrics for all evaluated models. As expected, strong in-dataset performance is observed across all evaluated models, reflecting the controlled training and testing conditions.

The FlowTransformer-lite baseline performs comparably to the MLP in this setting, indicating that self-attention mechanisms do not provide a significant advantage when domain shift is absent. Overall, these results establish a strong reference baseline under closed-world assumptions for subsequent cross-dataset evaluations.

From a deployment perspective, this observation suggests that architectural complexity alone does not guarantee cross-domain robustness in the absence of domain-aware training or adaptation.

### 5.2. Cross-dataset generalization

As shown in Fig 3(c–d), cross-dataset evaluation results in a substantial degradation in both macro-averaged F1-score and attack-class recall, highlighting the pronounced impact of domain shift when models are applied outside the conditions under which they were trained.

Among the evaluated classifiers, the MLP exhibits the most pronounced performance degradation, particularly in attack-class recall, indicating limited robustness to distributional changes across datasets. Random Forest and XGBoost demonstrate relatively higher stability under transfer; however, their performance still degrades markedly, suggesting that decision boundaries learned from tabular NetFlow features remain sensitive to domain mismatch across network environments.

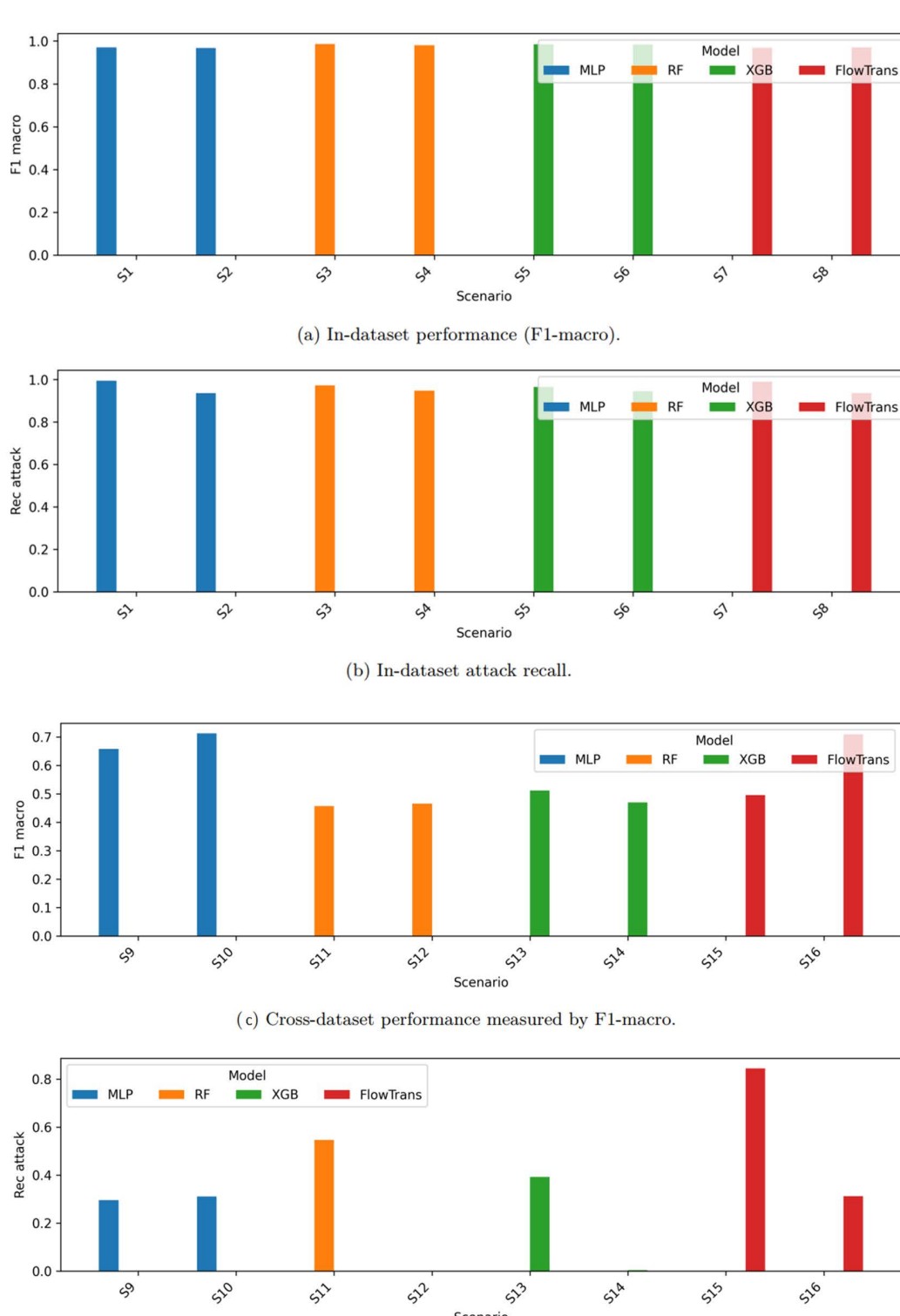

(a) In-dataset performance (F1-macro).

(b) In-dataset attack recall.

(c) Cross-dataset performance measured by F1-macro.

(d) Cross-dataset attack-class recall.

**Fig 3. Performance comparison across evaluation scenarios. (a–b)** In-dataset performance measured by F1-macro and attack recall. **(c–d)** Cross-dataset evaluation results. Preprocessing statistics are computed on training splits only to avoid data leakage.

**Table 7. In-dataset evaluation results (S1–S8).**

| ID | Scenario | Model | Train | Test | Acc | F1_macro | $Prec_{atk}$ | $Rec_{atk}$ | $N$ |
|----|----------|-------|-------|------|-----|----------|--------------|-------------|-----|
| S1 | MLP_UNSW→UNSW | MLP | UNSW | UNSW | 0.9861 | 0.9698 | 0.9046 | 0.9950 | 380,961 |
| S2 | MLP_CIC→CIC | MLP | CIC | CIC | 0.9861 | 0.9672 | 0.9480 | 0.9368 | 1,258,861 |
| S3 | RF_UNSW→UNSW | RF | UNSW | UNSW | 0.9940 | 0.9863 | 0.9786 | 0.9735 | 380,961 |
| S4 | RF_CIC→CIC | RF | CIC | CIC | 0.9917 | 0.9803 | 0.9831 | 0.9480 | 1,258,861 |
| S5 | XGB_UNSW→UNSW | XGB | UNSW | UNSW | 0.9933 | 0.9847 | 0.9807 | 0.9658 | 380,961 |
| S6 | XGB_CIC→CIC | XGB | CIC | CIC | 0.9932 | 0.9838 | 0.9986 | 0.9456 | 1,258,861 |
| S7 | FlowTrans_UNSW→UNSW | FlowTrans | UNSW | UNSW | 0.9855 | 0.9684 | 0.9035 | 0.9910 | 380,961 |
| S8 | FlowTrans_CIC→CIC | FlowTrans | CIC | CIC | 0.9875 | 0.9705 | 0.9597 | 0.9368 | 1,258,861 |

The FlowTransformer-lite baseline shows degradation patterns comparable to those of conventional models, indicating that the introduction of self-attention mechanisms alone is insufficient to address cross-dataset generalization when training is restricted to a single source domain. These observations reinforce that increased model capacity alone is insufficient to mitigate domain shift, thereby motivating the need for explicit domain-aware training strategies.

Table 8 reports detailed cross-dataset results for all evaluated scenarios. In several transfer directions, severe failure modes are observed. For example, when Random Forest is trained on NF-CSE-CIC-IDS2018 and tested on UNSW-NB15, the classifier collapses to predicting the benign class, yielding zero attack recall. Such outcomes underscore the limitations of closed-world training assumptions under realistic deployment conditions.

**5.2.1. Qualitative error analysis.** To better understand the observed recall degradation under cross-dataset transfer, we conducted a qualitative inspection of misclassified samples. Errors are frequently concentrated in regions where source- and target-domain flow distributions partially overlap, particularly for short-lived or low-volume attack flows whose statistical characteristics resemble benign traffic in the source domain. In addition, differences in traffic composition and flow-generation procedures across datasets shift the joint distribution of flow duration, packet counts, and byte rates, causing decision boundaries learned in the source domain to generalize poorly to attacks in the target domain.

Motivated by these severe degradation patterns, we next examine whether domain-aware training strategies, including mixed-domain training and lightweight fine-tuning, can partially recover performance under domain shift.

**Table 8. Cross-dataset evaluation results (S9–S16).**

| ID | Scenario | Model | Train | Test | Acc | F1_macro | $Prec_{atk}$ | $Rec_{atk}$ | $N$ |
|----|----------|-------|-------|------|-----|----------|--------------|-------------|-----|
| S9 | MLP_UNSW→CIC | MLP | UNSW | CIC | 0.8837 | 0.6587 | 0.5387 | 0.2955 | 1,258,861 |
| S10 | MLP_CIC→UNSW | MLP | CIC | UNSW | 0.9128 | 0.7130 | 1.0000 | 0.3103 | 380,961 |
| S11 | RF_UNSW→CIC | RF | UNSW | CIC | 0.5535 | 0.4574 | 0.1449 | 0.5462 | 1,258,861 |
| S12 | RF_CIC→UNSW | RF | CIC | UNSW | 0.8712 | 0.4656 | 0.0000 | 0.0000 | 380,961 |
| S13 | XGB_UNSW→CIC | XGB | UNSW | CIC | 0.6779 | 0.5123 | 0.1609 | 0.3921 | 1,258,861 |
| S14 | XGB_CIC→UNSW | XGB | CIC | UNSW | 0.8739 | 0.4698 | 0.9825 | 0.0035 | 380,961 |
| S15 | FlowTrans_UNSW→CIC | FlowTrans | UNSW | CIC | 0.5585 | 0.4955 | 0.1952 | 0.8441 | 1,258,861 |
| S16 | FlowTrans_CIC→UNSW | FlowTrans | CIC | UNSW | 0.9102 | 0.7096 | 0.9338 | 0.3125 | 380,961 |

## 5.3. Mixed-domain Training

As shown in Fig 4(a–b), mixed-domain training leads to consistent improvements in both macro-averaged F1-score and attack-class recall compared to direct cross-dataset evaluation. The observed asymmetry reflects differences in dataset scale, traffic composition, and flow-generation mechanisms across domains.

These results indicate that exposure to heterogeneous traffic distributions across multiple network environments during training helps models learn more domain-invariant decision boundaries. The performance gains are particularly evident for the MLP and FlowTransformer-lite models, which benefit from the increased diversity of training samples when compared to single-source training.

Nevertheless, mixed-domain training does not fully close the gap to in-dataset performance. This suggests that while aggregating data from multiple domains mitigates the effects of domain shift, residual discrepancies in traffic characteristics, data collection procedures, and flow statistics across datasets remain challenging to reconcile. Table 9 reports detailed mixed-domain training results for all evaluated models, complementing the trends observed in Fig 4(a–b).

The mixed-domain results should be interpreted with caution. The strong performance observed under this setting represents an optimistic upper bound enabled by large-scale exposure to heterogeneous traffic, rather than a default or always-realistic deployment configuration. In particular, the dominance of NF-CSE-CIC-IDS2018 in the mixed-domain training set likely contributes to the strong performance of tree-based models, especially under the binary classification formulation adopted in this study, where majority-class dominance can amplify the influence of large-scale datasets.

## 5.4. Fine-tuning with Limited Target Data

As shown in Fig 4(c–d), transfer-aware fine-tuning substantially improves attack-class recall across most cross-dataset transfer directions, demonstrating its effectiveness as a lightweight adaptation mechanism under domain shift.

Taken together, the results indicate that fine-tuning with a small amount of labeled target-domain data can significantly recover performance lost under direct cross-dataset transfer. This setting closely reflects realistic deployment conditions, where limited labeled samples from a new environment may be available, but full retraining on large-scale target data is impractical.

The largest performance gains are typically observed for the MLP and FlowTransformer-lite models, reflecting their greater capacity for representation adaptation when exposed to even limited target-domain supervision. Tree-based models (Random Forest and XGBoost) also benefit from fine-tuning, although their relative gains are generally smaller, as these models already exhibit stronger robustness under mixed-domain training.

Notably, fine-tuning effectiveness remains asymmetric across transfer directions. In particular, adapting models from NF-CSE-CIC-IDS2018 to UNSW-NB15 remains challenging for the MLP under the fixed 5% budget, with only marginal recall improvement. This behavior suggests that limited target supervision may be insufficient when the target domain exhibits more compact traffic distributions or reduced diversity in the unified NetFlow feature space. Table 10 reports detailed fine-tuning results obtained using a fixed 5% labeled subset of the target-domain training split, while validation and test sets remain unchanged. These results complement the trends shown in Fig 4(c–d).

Overall, these findings demonstrate that transfer-aware fine-tuning can substantially mitigate the impact of domain shift without requiring full retraining from scratch. However, residual performance gaps persist in challenging transfer directions, highlighting the inherent difficulty of cross-domain intrusion detection under constrained supervision in realistic deployment settings

## 5.5. Attack-level precision and recall analysis

Fig 5 compares attack-class precision and recall under cross-dataset transfer and transfer-aware fine-tuning, highlighting, through an attack-level evaluation lens, the risk-sensitive trade-off between missed attacks (false negatives) and false alarms (false positives) that arises when models are deployed under domain shift.

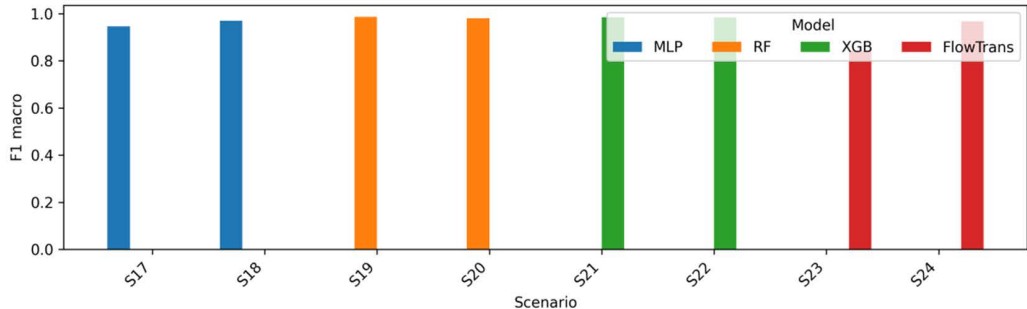

(a) Mixed-domain performance measured by F1-macro.

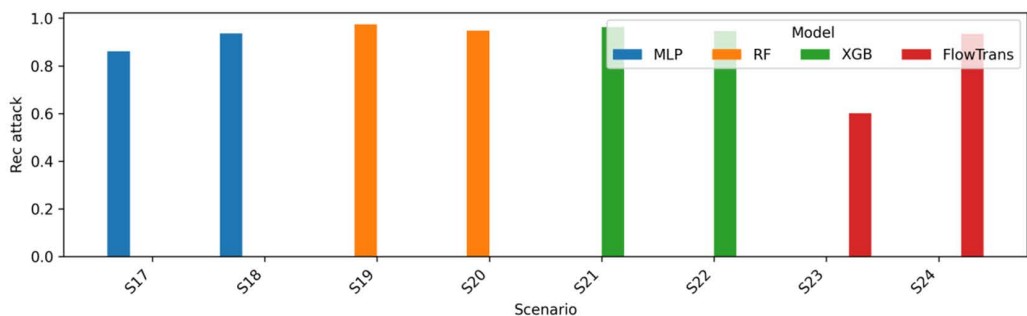

(b) Mixed-domain attack-class recall.

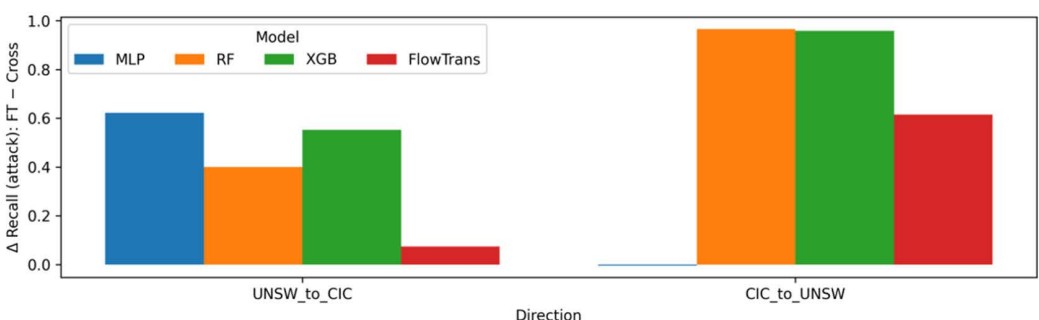

(c) Change in attack-class recall (ΔRecall) induced by fine-tuning under cross-dataset directions.

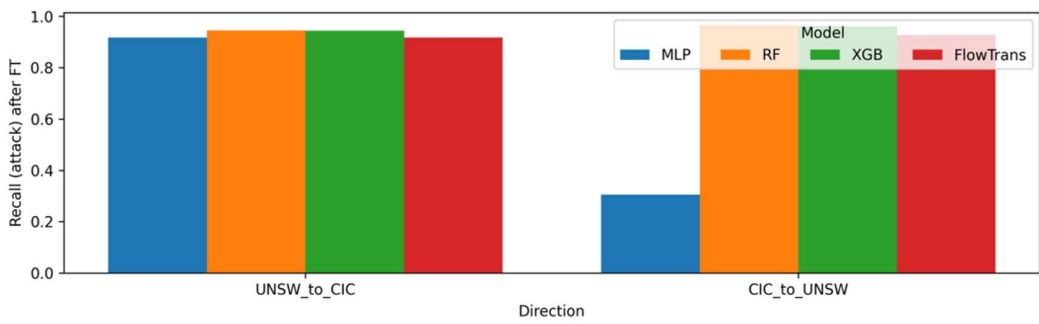

(d) Attack-class recall after fine-tuning.

**Fig 4. Transfer and adaptation performance across domains. (a–b)** Mixed-domain training results measured by F1-macro and attack recall. **(c–d)** Fine-tuning with limited labeled target-domain data.

**Table 9. Mixed-domain training results (S17–S24).**

| ID | Scenario | Model | Train | Test | Acc | F1_macro | Prec$_{atk}$ | Rec$_{atk}$ | N |
|---|---|---|---|---|---|---|---|---|---|
| S17 | MLP_MIX→UNSW | MLP | MIX | UNSW | 0.9772 | 0.9462 | 0.9543 | 0.8611 | 380,961 |
| S18 | MLP_MIX→CIC | MLP | MIX | CIC | 0.9876 | 0.9705 | 0.9603 | 0.9363 | 1,258,861 |
| S19 | RF_MIX→UNSW | RF | MIX | UNSW | 0.9939 | 0.9863 | 0.9783 | 0.9737 | 380,961 |
| S20 | RF_MIX→CIC | RF | MIX | CIC | 0.9917 | 0.9802 | 0.9829 | 0.9480 | 1,258,861 |
| S21 | XGB_MIX→UNSW | XGB | MIX | UNSW | 0.9933 | 0.9846 | 0.9843 | 0.9622 | 380,961 |
| S22 | XGB_MIX→CIC | XGB | MIX | CIC | 0.9933 | 0.9840 | 0.9996 | 0.9456 | 1,258,861 |
| S23 | FlowTrans_MIX→UNSW | FlowTrans | MIX | UNSW | 0.9401 | 0.8419 | 0.8895 | 0.6010 | 380,961 |
| S24 | FlowTrans_MIX→CIC | FlowTrans | MIX | CIC | 0.9861 | 0.9672 | 0.9496 | 0.9350 | 1,258,861 |

**Table 10. Fine-tuning results with 5% labeled target subset (S25–S32).**

| ID | Scenario | Model | Train | Test | Acc | F1_macro | Prec$_{atk}$ | Rec$_{atk}$ | N |
|---|---|---|---|---|---|---|---|---|---|
| S25 | MLP_FT_UNSW→CIC (5%) | MLP | UNSW | CIC | 0.9556 | 0.9041 | 0.7642 | 0.9174 | 1,258,861 |
| S26 | MLP_FT_CIC→UNSW (5%) | MLP | CIC | UNSW | 0.8708 | 0.6511 | 0.4831 | 0.3055 | 380,961 |
| S27 | RF_FT_UNSW→CIC (5%) | RF | UNSW | CIC | 0.9920 | 0.9809 | 0.9882 | 0.9453 | 1,258,861 |
| S28 | RF_FT_CIC→UNSW (5%) | RF | CIC | UNSW | 0.9922 | 0.9822 | 0.9717 | 0.9662 | 380,961 |
| S29 | XGB_FT_UNSW→CIC (5%) | XGB | UNSW | CIC | 0.9929 | 0.9830 | 0.9972 | 0.9442 | 1,258,861 |
| S30 | XGB_FT_CIC→UNSW (5%) | XGB | CIC | UNSW | 0.9919 | 0.9815 | 0.9734 | 0.9620 | 380,961 |
| S31 | FlowTrans_FT_UNSW→CIC (5%) | FlowTrans | UNSW | CIC | 0.9822 | 0.9580 | 0.9352 | 0.9173 | 1,258,861 |
| S32 | FlowTrans_FT_CIC→UNSW (5%) | FlowTrans | CIC | UNSW | 0.9806 | 0.9562 | 0.9202 | 0.9269 | 380,961 |

Relative to direct cross-dataset evaluation, both mixed-domain training and transfer-aware fine-tuning substantially improve attack recall while preserving acceptable precision levels. This balance is particularly critical for intrusion detection systems, where false negatives typically incur significantly higher operational risk than false positives.

Overall, these results emphasize that domain-aware training strategies not only recover detection performance under domain shift, but also enable more deployment-relevant precision–recall trade-offs at the attack level.

## 5.6. Discussion

The experimental results yield several key insights that are only observable through the transfer-aware evaluation design of TAN-IDS. First, strong in-dataset performance does not translate into cross-dataset robustness, highlighting a fundamental limitation of single-dataset evaluation protocols that dominate much of the IDS literature. Models that appear highly accurate under closed-world assumptions may suffer severe performance degradation when deployed in unseen network environments.

Second, architectural complexity alone is insufficient to address domain shift. Even models equipped with self-attention mechanisms, such as Transformer-based architectures, remain vulnerable when trained on a single source domain. This observation is consistent with prior findings on model uncertainty and calibration under distributional mismatch, where performance degradation persists despite increased model expressiveness [55].

Third, training strategies that explicitly incorporate domain awareness—namely mixed-domain training and transfer-aware fine-tuning—consistently outperform single-domain training across all evaluated models. These strategies mitigate performance collapse not by increasing model capacity, but by exposing classifiers to heterogeneous traffic distributions or limited target-domain supervision. This suggests that robustness to domain shift is primarily a data and evaluation problem, rather than a purely architectural one.

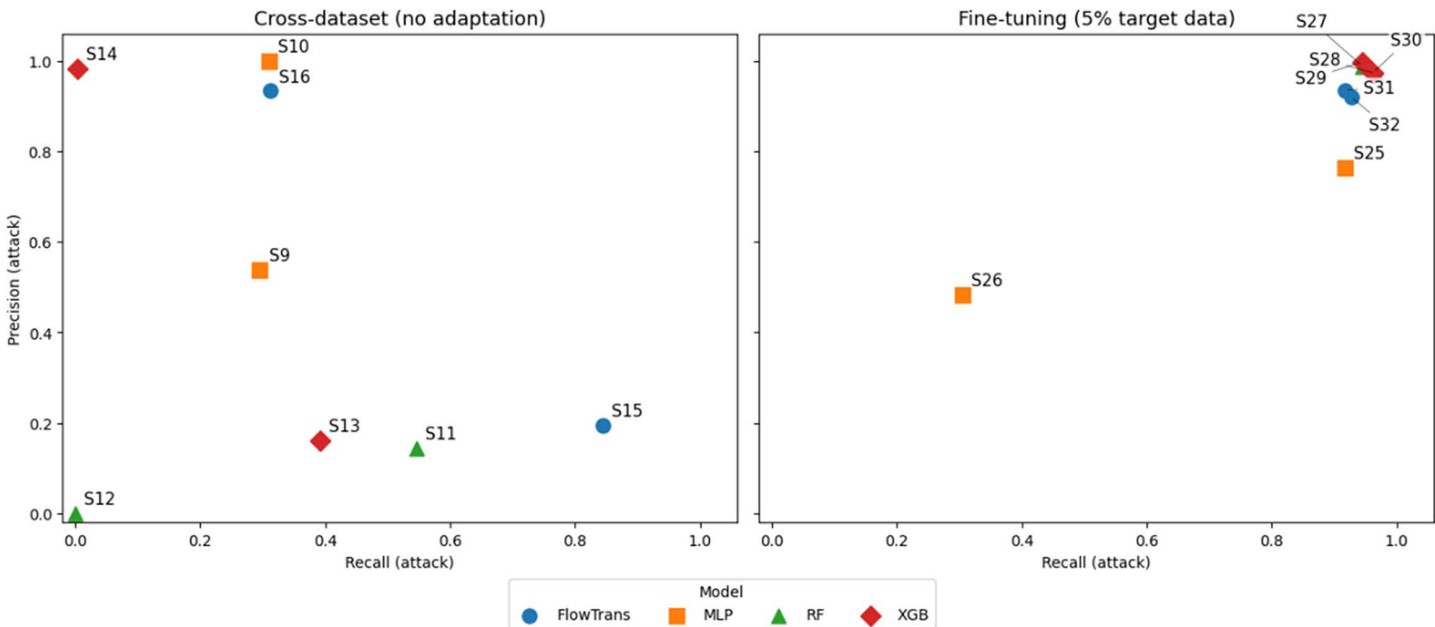

**Fig 5. Attack-class precision–recall across cross-dataset and fine-tuning scenarios.** The results illustrate how domain mismatch and limited target-domain adaptation jointly affect attack detection sensitivity and false-alarm propensity.

Taken together, these findings underscore the need to move IDS evaluation beyond benchmark-centric, closed-world settings toward deployment-oriented, domain-aware assessment. By combining a unified NetFlow feature interface with scenario-driven evaluation, TAN-IDS provides a reproducible and model-agnostic methodology for systematically studying performance degradation and recovery under realistic domain shift conditions.

**5.6.1. On the 8-feature NetFlow representation.** A central design decision in TAN-IDS is the adoption of a compact 8-dimensional NetFlow feature space shared across heterogeneous datasets. This representation is not intended to capture fine-grained attack semantics, such as multi-stage APT behaviors or protocol-specific exploit signatures. Instead, it is deliberately designed to assess whether intrusion detection models can robustly separate benign and malicious traffic under domain shift using deployment-realistic, flow-level signals that are commonly available in operational NetFlow/IPFIX monitoring environments.

By constraining the feature space, TAN-IDS explicitly prioritizes interoperability and reproducibility across datasets over maximizing in-dataset detection accuracy. This design choice reduces confounding effects introduced by dataset-specific, high-dimensional, or proprietary feature sets, thereby allowing observed performance degradation to be more confidently attributed to domain shift rather than feature-space incompatibility. Although richer feature representations may yield higher absolute performance within individual datasets, they are often impractical, unstable across collection environments, or unavailable in real-world deployments.

Accordingly, TAN-IDS focuses on binary attack detection within a unified flow-level interface as a first-order robustness assessment. Extending the framework to multi-class or attack-family discrimination is an important direction for future work, but is intentionally left out of scope to avoid conflating transferability analysis with dataset-specific label definitions. Similarly, a systematic feature ablation study is deferred, as the primary contribution of TAN-IDS lies in evaluation protocol design rather than feature engineering or optimization.

**5.6.2. Cross-dataset generalization.** Across all evaluated models, substantial performance degradation is observed when transferring from one dataset to another without any form of adaptation. This behavior is consistent with prior

large-scale cross-evaluation studies and reflects the pronounced domain shift between UNSW-NB15 and NF-CSE-CIC-IDS2018. Rather than indicating intrinsic model deficiencies, this degradation exposes the fundamental limitations of closed-world evaluation protocols and confirms that strong in-dataset performance does not reliably translate into robustness under real-world deployment conditions.

The observed collapse in attack recall under cross-dataset transfer can be attributed to feature distribution shift and dataset-specific collection biases. Even within a unified NetFlow feature interface, the joint distribution of flow-level statistics—including duration, packet counts, byte volumes, and rate-based attributes—varies substantially across datasets due to differences in network environments, background traffic composition, and attack generation procedures. Consequently, decision boundaries learned on the source domain may fail to generalize to target-domain attacks whose flow signatures deviate from source-domain patterns, leading to severe detection failures in unseen environments.

### 5.6.3. Model-specific behavior under domain shift.

The cross-dataset evaluation reveals distinct generalization behaviors across model families, with particularly pronounced differences observed for the MLP and FlowTransformer-lite architectures. In the CIC→UNSW transfer direction, the MLP suffers a severe collapse in attack recall, which persists even after transfer-aware fine-tuning with a limited amount of target-domain data. This behavior indicates that the MLP tends to learn highly source-dependent decision boundaries that are tightly coupled to the feature distributions of the training domain.

As a global parametric model, the MLP may encode strong correlations among flow-level features that do not remain stable when the underlying data distribution shifts across network environments. Under such conditions, the model can maintain high precision by conservatively predicting the benign class, but fails to generalize to previously unseen attack patterns, resulting in poor recall under domain mismatch.

Moreover, fine-tuning with a small fraction of labeled target-domain samples is not always sufficient to substantially reshape these global decision boundaries, particularly in transfer directions characterized by strong distributional divergence and class imbalance. This observation is consistent with prior findings that neural networks applied to tabular data can be especially sensitive to dataset shift, even when feature definitions remain unchanged but their joint distributions vary.

In contrast, FlowTransformer-lite exhibits moderately improved robustness in some cross-dataset settings. Although it is not explicitly designed for domain adaptation, its self-attention mechanism enables contextual aggregation across flow features, which may reduce sensitivity to feature-wise distribution shifts compared to pointwise MLP mappings. Nevertheless, FlowTransformer-lite still experiences substantial performance degradation under direct dataset transfer, reinforcing the conclusion that architectural complexity alone is insufficient to guarantee robust cross-domain generalization.

### 5.6.4. Effect of domain-aware training.

When trained using mixed-domain data, all evaluated models demonstrate improved robustness compared to direct cross-dataset evaluation. This result suggests that exposure to heterogeneous traffic distributions during training partially mitigates domain shift by encouraging models to rely on more stable and domain-invariant flow-level characteristics, rather than dataset-specific patterns.

In some settings, mixed-domain training yields pronounced performance gains. For example, the Random Forest model evaluated on UNSW-NB15 (S17) achieves an attack-class F1-score that exceeds its corresponding single-domain training baseline. This behavior does not indicate data leakage, as training and test splits remain strictly disjoint across all evaluation scenarios. Instead, the observed improvement can be attributed to increased training diversity and sample volume, which expose the model to a broader range of flow-level variations and decision boundary configurations.

More importantly, transfer-aware fine-tuning with a small fraction of labeled target-domain data leads to substantial recovery in attack recall and macro-F1 across all evaluated models. These results indicate that lightweight adaptation strategies are both practical and effective in deployment scenarios where only limited labeled traffic from new environments is available. Together, mixed-domain training and fine-tuning represent complementary forms of domain awareness: the former improves general robustness through data diversity, while the latter enables targeted adaptation to residual domain-specific discrepancies.

**5.6.5. Deployment implications.** Although cross-dataset performance is consistently lower than in-dataset results, this behavior is both expected and consistent with realistic deployment conditions. Unlike benchmark-oriented studies that report near-perfect accuracy under closed-world assumptions, TAN-IDS explicitly exposes domain shift and evaluates performance degradation and recovery strategies under a unified, deployment-realistic flow-level representation. This design prioritizes reliability, transparency, and reproducibility over optimistic performance reporting, making the evaluation more representative of operational intrusion detection systems.

From a deployment perspective, the primary benefit of transfer-aware fine-tuning lies in reducing labeling effort rather than optimizing computational efficiency. In this study, fine-tuning is performed using a small fixed budget (5%) of labeled target-domain data, which significantly limits annotation cost while avoiding full retraining from scratch. We do not explicitly profile inference latency, throughput, or hardware-specific resource consumption, and therefore make no quantitative claims regarding computational efficiency. A detailed analysis of system-level performance under real-time or streaming deployment constraints is consequently left for future work.

## 6. Threats to validity

This section discusses potential threats to the validity of the experimental results and the conclusions drawn from the TAN-IDS evaluation framework. Following common empirical research practice, we consider construct validity, internal validity, external validity, and reproducibility.

### 6.1. Construct validity

Construct validity concerns whether the experimental design and evaluation metrics appropriately capture the intended research objectives. In this work, intrusion detection is formulated as a binary flow-level classification task, in which all attack categories are aggregated into a single *attack* class and normal traffic is labeled as *benign*. This formulation is an intentional design choice aligned with the primary objective of TAN-IDS, namely to evaluate cross-domain robustness under domain shift rather than fine-grained attack categorization.

Multi-class evaluation across heterogeneous IDS benchmarks is complicated by inconsistent attack taxonomies, label granularity, and semantic mismatches.

Such discrepancies can introduce confounding factors and obscure whether observed performance changes stem from true distribution shift or dataset-specific labeling artifacts. The binary formulation therefore provides a controlled and comparable evaluation setting for cross-dataset analysis. While multi-class detection remains important for operational use, extending TAN-IDS to consistently aligned multi-class settings is left for future work.

A related construct validity consideration concerns the use of a compact 8-dimensional NetFlow feature space. Although richer, dataset-specific feature sets may yield higher absolute detection performance and may better capture complex or application-layer attack behaviors. However, such representations are often incompatible across datasets and less suitable for cross-platform deployment monitoring. The selected feature space therefore reflects a deliberate trade-off between expressive power and interoperability, consistent with the framework's focus on deployment-oriented evaluation rather than feature optimization.

### 6.2. Internal Validity

Internal validity relates to whether observed performance differences can be attributed to domain shift and training strategies rather than uncontrolled experimental factors. To mitigate this threat, all models are trained and evaluated using identical preprocessing pipelines, consistent data splits, and fixed hyperparameter configurations across all evaluation scenarios. This design ensures that performance differences primarily reflect changes in training data composition and deployment scenario rather than implementation variability.

Nevertheless, stochastic elements inherent to machine learning, particularly in neural network optimization, may introduce minor variance. While random seeds are fixed wherever possible, some nondeterminism may persist due to underlying software frameworks and hardware execution.

Fine-tuning results obtained under limited target-label budgets may also be sensitive to the specific subset of target-domain samples selected for adaptation. This variability is not explicitly quantified in the current study and reflects realistic deployment conditions, where labeled target data are scarce and often collected opportunistically. Accordingly, results are reported for single experimental runs with fixed random seeds, without confidence intervals or statistical significance testing.

### 6.3. External validity

External validity concerns the generalizability of the findings beyond the evaluated datasets. TAN-IDS is instantiated using UNSW-NB15 and NF-CSE-CIC-IDS2018, two widely adopted and heterogeneous benchmarks that capture distinct traffic characteristics and attack distributions. While these datasets provide a meaningful testbed for cross-domain evaluation, they do not represent the full diversity of real-world network environments.

Although the experimental evaluation focuses on these two benchmarks, the TAN-IDS framework itself is not tied to specific datasets and can be applied to additional flow-based IDS datasets. These benchmarks were selected because they represent heterogeneous and widely used IDS datasets with distinct traffic generation processes and attack distributions. Future work will extend the evaluation to additional flow-based datasets to further assess the generalizability of the framework.

Similar limitations have been reported for intrusion detection datasets in other domains, including IoT and IIoT settings [16,17]. As a result, absolute performance values and recovery trends observed in this study may differ when applying TAN-IDS to additional datasets or operational networks.

Furthermore, the experiments focus on offline flow-level analysis. System-level behavior under real-time deployment constraints is not explicitly evaluated. Such constraints include latency budgets, concept drift, and evolving traffic patterns. These aspects represent important directions for future work when integrating the framework into operational intrusion detection pipelines.

Mixed-domain training in this study follows the natural sample proportions of the constituent datasets, which may lead to dominance by the larger domain. While this setting reflects realistic large-scale monitoring environments with imbalanced data availability, it may overestimate robustness relative to balanced-domain mixtures or strictly isolated deployment scenarios.

Finally, TAN-IDS focuses on robustness to domain shift. Robustness against adversarially manipulated inputs represents an orthogonal deployment concern for machine learning-based IDS and is outside the scope of the present evaluation framework [56,57].

### 6.4. Reproducibility and Implementation Validity

Reproducibility is a central design objective of the TAN-IDS framework. All experiments follow an artifact-based evaluation workflow in which intermediate datasets, preprocessing configurations, trained model checkpoints, prediction outputs, and evaluation tables are persistently stored. This design enables experiments to be replicated, audited, and extended without requiring retraining from scratch, thereby supporting transparent and repeatable cross-domain evaluation.

To further promote implementation validity, all models are trained using fixed data splits, consistent preprocessing pipelines, and standardized hyperparameter configurations across evaluation scenarios. The scenario controller operates independently of the classifier implementation, ensuring that changes in performance can be attributed to evaluation conditions rather than uncontrolled implementation differences.

Nevertheless, absolute performance values may still be influenced by implementation-level factors such as software versions, hardware architectures, parallelization strategies, and low-level library optimizations. While these factors do not affect the qualitative conclusions drawn in this work, they may introduce minor numerical variability across platforms.

Future work will explore containerized execution environments, repeated multi-run evaluation, and statistical significance analysis to further strengthen reproducibility guarantees. These practices align with established recommendations for reproducible research in applied machine learning and data-driven systems [58,59].

## 7. Conclusion and future work

This paper introduced **TAN-IDS**, a transfer-aware evaluation framework designed to enable realistic, reproducible, and deployment-oriented assessment of flow-based intrusion detection systems under domain shift. Rather than proposing a new detection model, TAN-IDS addresses a critical gap in the IDS literature by focusing on evaluation methodology and benchmarking artifacts that bridge the disconnect between laboratory performance and real-world deployment conditions.

By mapping heterogeneous IDS datasets into a unified 8-dimensional NetFlow feature space, TAN-IDS minimizes feature-space incompatibility and enables direct, controlled cross-domain comparison of detection models. Extensive experiments on UNSW-NB15 and NF-CSE-CIC-IDS2018 demonstrate that near-perfect in-dataset performance can degrade substantially under direct dataset transfer, confirming that closed-world evaluation protocols often provide an overly optimistic and misleading view of IDS effectiveness.

The results further show that explicitly domain-aware training strategies are consistently more effective than increasing model complexity in isolation. Both mixed-domain training and fine-tuning with a limited amount of labeled target-domain data substantially mitigate performance degradation across all evaluated models, including classical ensembles, neural networks, and a lightweight Transformer-based baseline. These findings highlight practical adaptation mechanisms that are feasible in operational settings, where large-scale labeling of traffic from new environments is rarely possible.

Beyond quantitative performance metrics, TAN-IDS provides a structured lens for diagnosing cross-domain failures and understanding model behavior under domain shift. By organizing evaluation into explicit domain-aware scenarios under a deployment-realistic flow-level representation, the framework supports systematic analysis of robustness, transferability, and failure modes in machine learning–based intrusion detection.

Future work will extend TAN-IDS along several directions. First, incorporating additional datasets and traffic sources, including IoT, IIoT, and telemetry-oriented benchmarks, will strengthen external validity and broaden applicability. Second, extending the framework to consistently aligned multi-class and hierarchical attack classification settings may enable deeper analysis of fine-grained detection behavior under domain shift. Third, integrating online and continual learning mechanisms will allow TAN-IDS to support evaluation under evolving traffic patterns and streaming constraints. Finally, closer alignment with standardization efforts and operational monitoring interfaces will position TAN-IDS as a reusable evaluation artifact for the design, benchmarking, and deployment of robust flow-based cyber defense systems.

Evaluating expanded feature representations, when consistently available across datasets, represents an important direction for future work but remains orthogonal to the scenario-centric evaluation objective of TAN-IDS.

Overall, TAN-IDS reinforces the need to move beyond closed-world benchmarking and toward domain-aware evaluation protocols that more accurately reflect the challenges faced by real-world intrusion detection systems. These conclusions are consistent with prior large-scale cross-evaluation studies on IDS generalization [7].

## Author contributions

**Conceptualization:** Dung Ha Thanh.

**Data curation:** Dung Ha Thanh.

**Formal analysis:** Dung Ha Thanh.

**Investigation:** Dung Ha Thanh.

**Methodology:** Dung Ha Thanh.

**Resources:** Dung Ha Thanh.

**Software:** Dung Ha Thanh.

**Validation:** Dung Ha Thanh.

**Visualization:** Dung Ha Thanh.

**Writing – original draft:** Dung Ha Thanh.

**Writing – review & editing:** Dung Ha Thanh.

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
