## [Decision Letter · Decision Letter 0]

11 Mar 2026

Dear Dr. Ha Thanh,

Thank you for submitting your manuscript to PLOS ONE. After careful consideration, we feel that it has merit but does not fully meet PLOS ONE’s publication criteria as it currently stands. Therefore, we invite you to submit a revised version of the manuscript that addresses the points raised during the review process.

If applicable, we recommend that you deposit your laboratory protocols in protocols.io to enhance the reproducibility of your results. Protocols.io assigns your protocol its own identifier (DOI) so that it can be cited independently in the future. For instructions see: https://journals.plos.org/plosone/s/submission-guidelines#loc-laboratory-protocols. Additionally, PLOS ONE offers an option for publishing peer-reviewed Lab Protocol articles, which describe protocols hosted on protocols.io. Read more information on sharing protocols at. Additionally, PLOS ONE offers an option for publishing peer-reviewed Lab Protocol articles, which describe protocols hosted on protocols.io. Read more information on sharing protocols at . Additionally, PLOS ONE offers an option for publishing peer-reviewed Lab Protocol articles, which describe protocols hosted on protocols.io. Read more information on sharing protocols at . Additionally, PLOS ONE offers an option for publishing peer-reviewed Lab Protocol articles, which describe protocols hosted on protocols.io. Read more information on sharing protocols at https://plos.org/protocols?utm_medium=editorial-email&utm_source=authorletters&utm_campaign=protocols....

We look forward to receiving your revised manuscript.

Kind regards,

Zeheng Wang

Academic Editor

PLOS One

**Journal Requirements:**

https://journals.plos.org/plosone/s/file?id=wjVg/PLOSOne_formatting_sample_main_body.pdf andandandand

4. Please upload a new copy of Figures 1, 2, 3, 4, 5, 6, and 7 as the detail is not clear. Please follow the link for more information:  https://journals.plos.org/plosone/s/figures

Reviewers' comments:

Reviewer's Responses to Questions

**Comments to the Author**

1. Is the manuscript technically sound, and do the data support the conclusions?

Reviewer #1: Partly

Reviewer #2: Partly

2. Has the statistical analysis been performed appropriately and rigorously?

Reviewer #1: Yes

Reviewer #2: I Don't Know

3. Have the authors made all data underlying the findings in their manuscript fully available?

Reviewer #1: Yes

Reviewer #2: No

4. Is the manuscript presented in an intelligible fashion and written in standard English?

Reviewer #1: Yes

Reviewer #2: Yes

Reviewer #1: 1)Sections 1.5, 1.6, and 1.7 are quite repetitive. The authors state multiple times that they are proposing an evaluation framework rather than a new model, and they repeatedly emphasize the unified NetFlow interface.

2)While the 8-dimensional representation enables excellent interoperability, it may risk stripping away the nuanced data required to detect highly complex or novel attacks (e.g., zero-day exploits, low-and-slow application-layer attacks). A brief acknowledgment of this trade-off—whether the 8 features represent a "lowest common denominator" that caps maximum detection capabilities—would strengthen the discussion.

3)The “RELATED WORK” section: it is suggested to add some updated literature about attack detection and machine learning techniques, such as:

• https://doi.org/10.1109/ACCESS.2025.3545918

• https://doi.org/10.32604/cmc.2023.041667

• https://doi.org/10.1109/ACCESS.2025.3543127

Reviewer #2: 1. The definition of “transfer-aware” is overly vague.

2. Using only two datasets results in insufficient external validity. Are the conclusions applicable only to UNSW and CIC?

3. Fine-tuning utilized only 5% of the data, with no ablation experiments conducted.

4. Why choose 8 dimensions? The rationale is insufficient.

5. Excessive charts with densely packed curves compromise readability. Merge Figures 3 to 7 into two larger figures.

6. Excessive use of long English sentences and high paragraph density

.

Reviewer #1: No

Reviewer #2: No

---

## [Author Response · Author response to Decision Letter 1]

14 Mar 2026

We thank the editor and reviewers for their valuable comments and suggestions. We have carefully revised the manuscript according to the reviewers’ feedback. A detailed point-by-point response to all comments is provided in the uploaded “Response to Reviewers” document, and all changes in the manuscript are highlighted in the Track Changes version.

---

## [Editor Report · Decision Letter 1]

24 Mar 2026

A transfer-aware, deployment-oriented evaluation framework for NetFlow-based intrusion detection systems (TAN-IDS)

PONE-D-26-04957R1

Dear Dr. Ha Thanh,

We’re pleased to inform you that your manuscript has been judged scientifically suitable for publication and will be formally accepted for publication once it meets all outstanding technical requirements.

Kind regards,

Zeheng Wang

Academic Editor

PLOS One
---

## [Editor Report · Acceptance letter]

PONE-D-26-04957R1

PLOS One

Dear Dr. Ha Thanh,

I'm pleased to inform you that your manuscript has been deemed suitable for publication in PLOS One. Congratulations! Your manuscript is now being handed over to our production team.

Kind regards,

on behalf of

Dr. Zeheng Wang

Academic Editor

PLOS One